# [Re] Graph Edit Networks

1 ## Reproducibility Summary

2 **Scope of Reproducibility**

3 The studied paper proposes a novel output layer for graph neural networks (the graph edit network - GEN). The objective
4 of this reproduction is to assess the possibility of its re-implementation in the Python programming language and the
5 adherence of the provided code to the methodology, described in the source material. Additionally, we rigorously
6 evaluate the functions used to create the synthetic data sets, on which the models are evaluated. Finally, we also pay
7 attention to the claim that the proposed architecture scales well to larger graphs.

8 **Methodology**

9 For most of our work, we were able to use the code, provided in the supplementary repository. We also offer our own
10 variations of the experimental setup, with an alternative method of risk estimation. A portion of the report is also
11 devoted to a more exhaustive description of the included data generating functions, otherwise not offered original paper.

12 **Results**

13 We were able to reproduce GEN's out-performance of a chosen baseline and its perfect scores on synthetic data sets.
14 We also confirm the author's claims of the sub-quadratic scaling of GEN's forward passes and deduce that they reported
15 the scaling of back-passes too favourably. We conclude our work with scepticism of the chosen experiments' suitability
16 to evaluate the model's performance and discuss our findings.

17 **What was easy**

18 All the provided code has extensive documentation which made the paper's experiments easy to reproduce. The entire
19 code base is readable, modular and adheres to established practices on code readability. The authors also provide some
20 unit tests for all of their models and have pre-implemented several useful diagnostic measures.

21 **What was difficult**

22 Running some of the provided code on a consumer-grade laptop (as reported in the original work) was prohibitively
23 expensive. The lack of transparency about the code base's runtimes made our work here much more difficult. Another
24 time-consuming task was the debugging of a section of author-provided code. We've helped the authors identify the
25 problem, which has now been resolved.

26 **Communication with original authors**

27 The authors were prompt with their responses, welcomed our efforts in reproducing their work and made themselves
28 available for any questions. Upon our request, they happily provided additional implementations, not originally available
29 in their repository, and offered their counter-arguments to some methodological concerns that we expressed to them.

# 1  Introduction

The studied paper proposes a novel output layer for graph neural networks (GNNs), the graph edit network (GEN). This layer yields a sequence of graph edits $\delta$ . Particularly, the graph edit schema considered in the work is the one initially proposed in [SF83], describing notions of node insertions ($\texttt{ins}_x$), deletions ($\texttt{del}_x$) and replacements ($\texttt{repl}_{i,x}$), as well as edge insertions ($\texttt{eins}_{i,j}$) and deletions ($\texttt{edel}_{i,j}$). Note that the subscripts $x$ in node edits refer to the attributes of the edited node (in $\texttt{repl}_{i,x}$, the additional subscript $i$ denotes the to-be replaced attributes), and $i, j$ in the edge edits refer to the indices of nodes between which the edited edge can be found.

These finite sequences of edits, also referred to as edit scripts $\bar{\delta}_t = [\delta_t^1, \delta_t^2, \ldots, \delta_t^n]$, are general enough to describe any graph-to-graph transformation and are not only very interpretable for humans, but also computationally efficient. Both of these properties establish GENs as a useful tool for work in the domain of graph time series prediction. More particularly, GENs perform time series prediction under the Markovian assumption, which states that knowing the graph $G_t$ and the mapping function $\psi_t$, derived from the edit script $\bar{\delta}_t$, is sufficient for predicting the graph found in the next step of the time series as

$$G_{t+1} = \psi_t(G_t); \quad \psi_t := \delta_t^1 \circ \delta_t^2 \circ \cdots \circ \delta_t^n; \quad \forall \delta_t^i \in \bar{\delta}_t,$$

where the subscript *t* denotes the time-dependant index in the time series.

# 2  Scope of Reproducibility

The authors of the reproduced work formally prove theorems, stating that finding a mapping $\psi$ between pairs of time-adjacent graphs is sufficient for constructing training data for GENs. They propose that their GNN architecture be trained to reproduce specific teaching signals for this function $\psi$, which may be derived from any gathered training time series of graphs. This is done by first finding reference pair mappings $\psi_t : G_t \to \bar{\delta}_t(G_t) \equiv G_t \to G_{t+1}$ from the training series via graph edit distance approximators[1], and then computing teaching signals via an algorithm, provided in the paper's supplementary material.

The authors empirically underpin this corollary by showing that the GEN performs well in a series of graph time-series prediction tests. They define several data generating processes (DGPs), from which the GEN attempts to learn the user-defined functions $\psi$, which remain hidden to the algorithm. The tests can be roughly split into three classes, which have corresponding experiments in section 4. The explicit conclusions of the experimental subsection of the original paper are that the GEN outperforms the selected baselines in all of the observed tasks.

In our work, we compare the GEN to one of the baselines - the modified version of Variational graph autoencoders (VGAE). As in the original work, we observe a modification of the method, suggested by [HHD+19], where the method attempts to directly infer the the graph in the next step of the time series. In the other experiments, we interpret claims about GEN's performance on different datasets directly.

Since the graphs, generated by the author-defined DGPs, are of a completely synthetic nature and very limited in scale, the authors also attempt to establish that GENs scale well to real-world networks. In their experimentation, they only pay attention to the scaling efficiency of the architecture and not to the quality of the predictions themselves. From the described conclusions, we identify the following claims, made in the experimental section of the paper, that we will be exploring:

**Claim** (i):   GENs, trained with either hinge or crossentropy loss, outperform the modified VGAE on all three dynamical graph system DGPs.

**Claim** (ii):   GENs, trained with user defined losses, achieve a perfect accuracy score on both dynamical tree DGPs.

**Claim** (iii):   The runtime of forward passes of a GEN, trained on the social network dataset (with or without edge filtering), scales sub-quadratically as the number of nodes in a graph increases.

**Claim** (iv):   The runtime of backward passes of a GEN, trained on the social network dataset with edge filtering, scales approximately linearly, as the number of nodes in a graph increases.

---

[1]Approximation is used due to the NP-hard nature of the graph edit distance in general, as shown in [BBC+17]. In practice, exploiting domain knowledge may also lead to sensible mappings $\psi_t$. As an example of domain knowledge exploitation, the authors cite [ZS89].

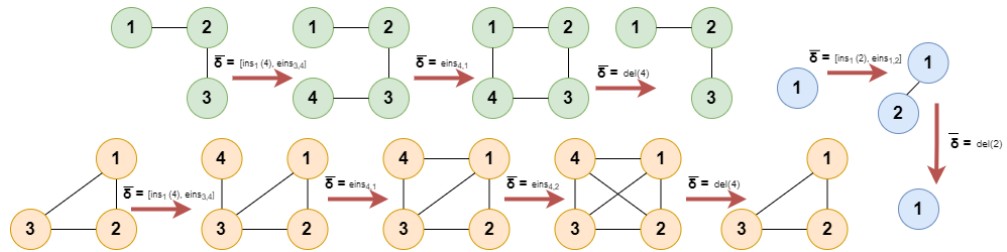

Figure 1: The three cyclical time series yielded by the Edit Cycles DGP.

An additional contribution of our work is the thorough study and description of the synthetic datasets, used to evaluate the GENs performance. We pay special attention to this part of the paper, as they were not exhaustively described in the original work. This examination helps us shed light on the performance of the GEN in the discussion section and evaluate the suitability of the used exprimental approachs. It also provides a more in-depth descriptive resource to other researchers in the field, that might find these DGPs useful for their own work.

# 3   Methodology

Throughout our reproduction attempt, we have made great use of the code, provided in a supplementary repository to the original paper [bpa21]. To replicate the author's experimental environment, we try to make the same assumptions and hyperparameter choices than those provided either in the original paper, or the documentation of the supplementary repository. A fork of this repository with our changes and additions is available at [Git].

## 3.1   Model descriptions

In the first class of experiments, we train 2 GEN models, one using the adapted cross-entropy loss (GEN-XE) and the other using the adapted hinge loss (GEN), described in the paper. Both models are parametrized by their input, output and hidden dimensionalities, as well as their used nonlinearities. Given the short edit scripts expected in these scenarios, no edge filtering is used in these models.

We also train the Variational Graph autoencoder model, as described in [KW16]. Apart from its input, output and hidden dimensionalities, it is also parametrized by the size of its encoding space, the regularization strength $\beta$ and a scaling factor for the noise on the last layer node features $\sigma$. It also takes a hyperparametric definition of the used nonlinearity.

The GEN models used in the experiments, governed by the Peano addition and Boolean formulae DGPs, are similar to those in the *Dynamical graph systems* class. The models here, however, use an author-defined loss function, with respect to a custom teaching protocol, with only a single predictive step between graphs. Similarly to before, no edge filtering is used.

In the experiments on the social network dataset, we train two variations of the GEN model. The first sets up two binary classifiers for each node to decide whether to consider changing outgoing/incoming edges or not. This approach is denoted in the results as *flexible edge filtering*. The second model limits the number of permitted edge edits with a fixed upper bound - this is denoted as *fixed edge filtering*. The models use a simplified single-step teaching protocol, over which its loss function is defined. In the protocol, all edits, except for node insertion, are processed as expected. For insertion, however, the protocol lets a given node $n$ insert a neighbor $n'$ when there is at least one edge *(n, n')* found in $G_{t+1}$, where $n'$ is not a node found in $G_t$. The authors acknowledge potential shortcomings of this method, but cite the desire of using a single-step protocol as the reason for choosing it.

## 3.2   Data

The paper contains three classes of experiments. The first two use user created DGPs, whereas the last one works with an external, well established social network. We describe the dataset and DGPs in accordance to the class of experiments they correspond to, in the following subsections.

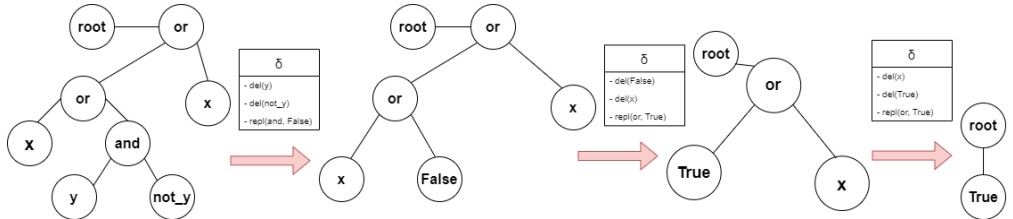

Figure 2: Example time series evolution of a graph, sampled from the Boolean Formulae DGP. The leftmost graph denotes the logical formula, $(x \vee (y \wedge \neg y)) \vee x$, whereas each evolution corresponds to a logical simplification of the previous graph.

### 3.2.1 Dynamical graph systems

The *Dynamical graph systems* class of DGPs governs the train and test set generation in Experiments 4.1.1, 4.2.1 and 4.2.2. The class contains three discrete processes, provided in the supplementary repository in the form of scripts for the python programming language. During training/testing time, the time series generator function is called, always returning a sequence of graphs based on DGP-specific function arguments.

The **Edit Cycles** DGP always yields one of three author specified cyclical time series, the outputs only differing in length and the starting time index. The edit script $\bar{\delta}_t$ between two graphs is always of cardinality $|\bar{\delta}| \leq 2$ and all possible generated graphs consist of between two and four nodes. The cyclical series that the DGP yields are visualized in Figure 1.

The **Degree Rules** data generating function generates a series of a determined length using the edit rules, described in Algorithm 1. The generator function accepts parameters, corresponding to the series length and the number of nodes in the initial graph $G_0$. $G_0$'s adjacency matrix is then randomly initialized. Consequentially, given a fixed time series length, the returned series is fully dependant on the random initialization of $G_0$, as the rules are deterministic. In the examples in section 4.1, as per the author's source code, the randomization from NumPy's `random.rand` is used, and all series' initial graphs $G_0$ start with exactly 8 nodes. We comment on this choice of randomization and provide our alternative in Section 4.2.

The third and final DGP in this class is inspired by Conway's **Game of Life** [Gar70]. Similarly to *Degree Rules*, it takes an input graph and applies a graph-to-graph mapping function. This one is specified by Algorithm 2 and is used to create a time-series of a specified length. This function is also deterministic. In the resulting graphs, the nodes considered *alive* in the Game of Life rule set are denoted with the feature value $x_n = 1$. In contrast to degree rules, Game of Life graphs retain their number of nodes throughout evolution, as the graph will always denote the $D \times D$ grid with the neighborhood structure modeling a nodes' 8-neighborhood, and only the nodes' alive/dead state will change. In each time series, a number of random Game of Life oscilators (randomly chosen between 5 candidates) is chosen and made alive. Afterwards, each still dead cell will be made alive with a probability $\Pr(\texttt{repl}_{0,1}(n)) = p$. In the experiments in section 4.1, we report results using the parameters $p = 0.1, D = 10$, and always placing a single oscillator on the grid at initialization.

### 3.2.2 Tree dynamical systems

The *Tree dynamical systems* class of DGPs governs the train and test set generation in Experiments 4.1.2 and 4.2.3. It contains two distinct processes. They are distinguished from the DGP class in the previous section because they both generate strictly tree-structured graphs, with no loops. Furthermore, they both include more complex node attribute encodings in the form of one-hot vectors.

The initial graph in a series, generated by the **Boolean Formulae** generator function, corresponds to a random Boolean formula. The time series following such a $G_0$ represents gradual simplifications of the formula, ending with a logic graph that can not be simplified any longer. An example evolution is given in Figure 2 for the formula $(x \vee (y \wedge \neg y)) \vee x$. The initial trees are generated via a stochastic regular tree grammar with a $\Pr(\wedge) = \Pr(\vee) = 0.3$ and $\Pr(x) = \Pr(\neg x) = \Pr(y) = \Pr(\neg y) = 0.1$. The generator functions also offer a hyperparametric maximal number of applied rules $p$, where the authors use $p = 3$ in the original experiments.

**Algorithm 1** The $G_t \to G_{t+1}$ mapping for the Degree rules DGP. The function shareN returns true if the nodes share at least one neighbor.

---

**Input:** Graph $G_t$, containing nodes $n$.
1: **for each** component $C \in G_t$ **do**
2:     **for each** $n \in C$ **do**
3:         $d \leftarrow$ degree$(n)$
4:         **if** $d \geq 3$ **then** del$(n)$
5:         **else if** $\exists n' \in C :$ shareN$(n, n')$ **then**
6:             **for each** $n' \in C :$ shareN$(n, n')$ **do**
7:                 eins$(n, n')$
8:         **else** ins$_1(n^*)$, eins$(n, n^*)$

**Algorithm 2** The $G_t \to G_{t+1}$ mapping for the Game of Life DGP. The AliveDegree function returns the number of neighboring nodes $n'$ with the attribute $x_{n'} = 1$.

---

**Input:** Graph $G_t$, containing nodes $n$.
1: **for each** $n \in G_t$ **do**
2:     $d \leftarrow$ AliveDegree$(n)$
3:     **if** $(x_n == 1)$ **and** $(d < 2$ **or** $4 \leq d)$ **then**
4:         repl$_{1,0}(n)$
5:     **else if** $(x_n == 0)$ **and** $(d == 3)$ **then**
6:         repl$_{0,1}(n)$

| | Graph Cycles | Degree Rules | Game of Life | Boolean Formulae | Peano Addition |
|---|---|---|---|---|---|
| **# of unique graphs** | 9 | 12346 | $2^{100}$ | 10788 | 34353 |

Table 1: The number of unique graphs that can appear in the time series, sampled from the DGPs in sections 3.2.1 and 3.2.2. as reported by the authors.

The **Peano addition** DGP models Peano's recursive definition of addition. The operations are encoded similarly as in the Boolean formulae DGP, where both the operands and the arguments are represented as nodes in the dynamical tree graph.The initial graph generator function receives an argument, specifying the maximal number $n$ of additions. The authors use $n = 3$ in their experiments. Peano's addition rules simplify into four edit rules, the edit scripts of which are all upper bound as $|\bar{\delta}| \leq 3$. The node attributes appearing in the set are the 10 digit values, the summation operation $+(m, n) = m + n$ and the successor operation $succ(m) = m + 1$.

The author-reported numbers of possible graphs, appearing in the time series, resultant from the five described DGPs, is tabulated in Table 1. Note, however, that not all of these graphs can be sampled as the initial graphs $G_0$ in a given series and that the mappings $\psi : G_t \to G_{t+1}$ are deterministic in all DGPs. Hence, the actual number of unique pairs $(G_t, G_{t+1})$ is much lower.

### 3.2.3 Real-world social network

For the final class of experiments, the arXiv HEP-Th citation network data set, first described in [LKF05], is used. It describes a graph, parsed from the e-print arXiv and covers all mutual citations within a set of 27,700 papers. In it, a paper $x$, that cites paper $y$ is connected with it with an outgoing edge. From this network, the authors parse sub-graphs with a rolling window approach - considering only papers published within $\tau$ months of a given time point between January 1993 to April 2003. The number of nodes naturally grows with $\tau$, so the result is a collection of graphs with different orders of node-count magnitude. In the presented experiment, these 1554 discovered sub-graphs of node count $N_G \in [100, 2786]$ are assumed as undirected.

### 3.3 Hyperparameters

For all the GNN-based models in the first two classes of experiments, the authors use two hidden layers with 64 neurons each. As far as the architecture specification is concerned, the GENs use summation as the aggregation function and concatenation as the merge function. All networks are trained with the Adam optimizer using the learning rate of $10^{-3}$. The weight decay is set to $10^{-5}$ in the graph dynamical systems class of experiments and to $10^{-3}$ in the dynamical tree class of experiments.

The results for the VGAE model are reported using $\beta = 10^{-3}, \gamma = 10^{-3}$. The dimensionality of its embedding space is always equal to the size of the last hidden layer, so 64. As per the provided code by the authors, all models use the sigmoid nonlinearity in the experiment on the *Game of Life* dataset, whereas we employ ReLU for all other experiments on synthetic data.

In the experiments reported in section 4.1, both the training and the testing time series are sampled independently from their corresponding DGP, without special assertions of training and testing set discrepancy. All models train on 30,000 series, whereas the testing results are reported for 10 samples. We comment on the authors' methods of risk estimation and provide alternatives for these parameters in section 4.2. For the experiments on the social network dataset, a 3-hidden layer architecture with the tanh nonlinearity, and PyTorch's default learning rate and weight decay are used.

### 3.4 Experimental setup and code

In our experiments, we use the metrics of precision and recall to evaluate the performance on insertion and deletion tasks. The experiments done on *Tree dynamical systems* use the notion of *accuracy*, which is an indicator function, defined at the value 1 when the nodes in the two input graphs match in all their features, and their adjacency matrices are identical. The reported *accuracy* is the average value of these indicator functions across all graph pairs in all time-series in the test set.

The experiments in section 4.1 were run in a loop across an entire class of DGPs, with 5 repetitions being ran for each considered model. In the training phase, a time series was independently generated on each epoch using its corresponding generator function. As per the original paper, the considered stopping criterion was a rolling 10-epoch average stop loss. Upon finishing training, the model was evaluated on time series, generated by the same generator functions as during training.

We recognized this method of risk estimation as potentially problematic, given that there is no special care taken to ensure the discrepancy of the tranining and testing sets. It is for this reason that we change the used approach in some experiments, reported in section 4.2. In them, we sample our test set of graphs $G_0^{\text{Test}}$ ahead of time, and ensure that at each sampled training time series, the function $\psi : G_0^{\text{Test}} \to G_1^{\text{Test}}$, $\forall\, G_0^{\text{Test}} \in T$ remains hidden from the algorithm.

### 3.5 Computational requirements

All experimentation was done on a desktop machine, running Windows 11, powered by an AMD Ryzen 7 2700X processor and 32 GB of RAM. The code was evaluated locally, in an environment, based on Python 3.8. The code base provided by the authors is dependant on the NumPy, PyTorch, PyTorch Geometric, Edist [PMH15] and MatPlotLib packages.

One repetition of running all three considered models on all three Graph dynamical systems (together) takes 90 minutes on average, with the VGAE taking the bulk of time to train, as the hinge-loss GEN usually hits the stop loss threshold and stops training earlier. A single repetition of the experiment on the Peano addition DGP takes approximately 15 minutes, whereas one over the Boolean formulae experiment takes 1 minute. On average, 60 minutes required to compute a full pass over all 12 months on the Social network experiment, for both edit schemas together. Working only with the largest graphs, i.e. $\tau = 12$, takes 8 minutes on average.

## 4 Experiments

Our results confirm the authors' findings from claims (i) - (iii) when considering the results of the strict reproduction. We find that the scaling of the backward passes from claim (iv) is not linear, but remains sub-quadratic. However, we show that these results are achieved by an architecture that is not able to optimize its loss function successfully.

Our additional experiments in Subsection 4.2 show that the experimental results are stable for different choices of the initial graph $G_0$. The results also stand for more robust method of risk estimation. From these additional experiments, we derive important insights about the testing scenarios, presented in Section 5.

### 4.1 Experiments reproducing original paper

#### 4.1.1 Precision/Recall on Dynamical Graph System DGPs

In this task, we aimed to reproduce the results, stated in Claim (i) in Section 2. For almost all the metrics, we were able to reproduce the values originally reported in the paper, with the difference $\delta := (\text{our results} - \text{reported results})$ within a standard deviation of 0. The only major discrepancy we noticed was an increase in mean deletion precision and

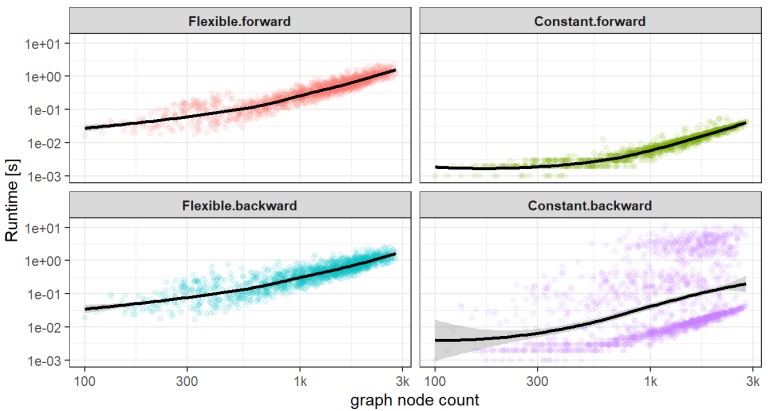

| Pass direction | Edge filtering | Log-log linear fit slope |
|---|---|---|
| **Forward** | Flexible | $1.38 \pm 0.02$ |
| | Constant | $1.31 \pm 0.02$ |
| **Backward** | Flexible | $1.30 \pm 0.01$ |
| | Constant | $1.69 \pm 0.10$ |

Figure 3: The runtime - graph scale dependence in the experiment 4.1.3, with overlaid fitted loess models. Each facet corresponds to an individual experiment, and the grey bands denote the 95% confidence interval of the fit.

Table 2: Slopes of log-log linear models on the Runtime/Graph scale scatter plot. The uncertainty denotes the standard deviation of slopes accross 5 repetitions of the experiment 4.1.3.

insertion recall for the VGAE model in the edit cycle task, when comparing to the results, reported in the original paper. However, both GEN models still outperformed the VGAE, which supports Claim (i).

### 4.1.2 Accuracy on Tree dynamical system DGPs

In this task, we address Claim (ii) from Section 2. In the original paper, the authors reported a 100% accuracy for both *Tree dynamical system* scenarios. While our results returned an accuracy of $0.98 \pm 0.02$ in the Boolean Formulae task (and a perfect score for Peano addition), we can conclude that these results are convincing enough to support Claim (ii).

### 4.1.3 Scaling of GENs on bigger graphs

This experiment addresss claims (iii) and (iv) from Section 2. In the original paper, the authors claim that GENs were able to scale sub-quadratically in their forward passes and approximately linearly in their backward passes, when using appropriate edge filtering approaches. Figure 3 shows scatter plots of the runtime-graph scale dependency on a log-log scale. Notice, that the runtime duration of the backward passes with constant edge filtering is very unstable, when compared to other scenarios. This is likely due to a higher difference in the fraction of considered edges, when compared to the flexible filtering approach. The scaling coefficients of the fitted linear models are further tabulated in Table 2. These results support Claim (iii) in that the forward passes scale sub-quadratically. However, the lower of the two average coefficients for computing the gradient (the flexible approach) is still substantially larger than one. This indicates an exponential, albeit sub-quadratic scaling of the backward passes. We conclude that these results do not support Claim (iv).

## 4.2 Experiments beyond original paper

### 4.2.1 Established methods of random graph generation

It is a common practice in the social network analysis (SNA) community to, when initializing random graphs, use specific methods of graph generation. Namely, if we want to make general statements about SNA methods, inferred from experiments on random graphs, these should be similar to those that tend to appear in nature. At the very least, it is considered a good practice to use established randomization methods, to more easily compare to results in other publications. In this experiment, we repeat the methods from experiment 4.1.1 on the Degree rules DGP. However, instead of randomly initializing the adjacency matrix, we use two established methods of random graph generation: the Erdős–Rényi model [ER59] and the Configuration graph model [New03]. In our experiment, graphs $G_0$ were always initialized with 36 nodes in both models. We set the edge creation probability in the Erdős-Renyi model $p = 0.5$ and the degree sequence of the Configuration model follows a random power-law sequence with the exponent $\gamma = 3$.

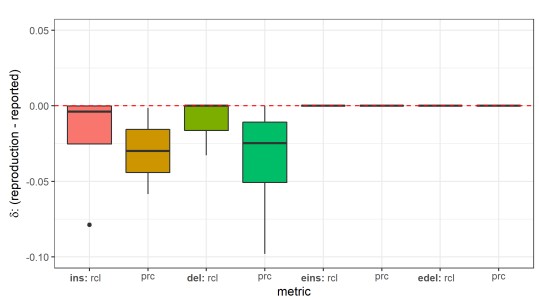

Figure 4: Diagnostic $\delta$-boxplot comparing the initially reported scores to our results using the alternative risk estimation method and a larger test set, performed on the Game of Life DGP for the hinge-loss based GEN.

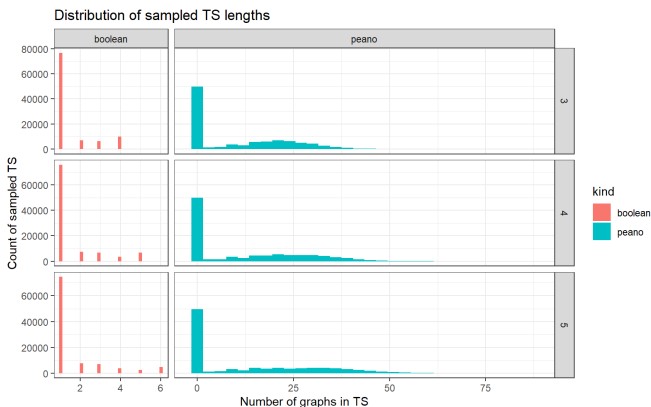

Figure 5: Distributions of time series lengths, sampled from the Tree Dynamic Systems DGPs. The facet rows correspond to the maximal number of operations (3-5).

All metrics on these newly generated random graphs remained in the 0.05-neighborhood of the originally reported results. We conclude that the performed experiments are robust to different methods of random graph generation, and that the change in graph generation does not disprove Claim (i).

### 4.2.2 Alternative methods of risk estimation - Dynamical graph systems

As established above, no special care is taken to ensure the discrepancy between the training and testing set of time-series in the original results. In this experiment, we re-run experiments 4.1.1 and 4.1.2 with our changed method of risk assessment, described in Section 3.4. We also raise the cardinality of testing set to 100, attempting to achieve stable results. We analyze our results by comparing several repetitions of the new experiment with the reported values. As a diagnostic tool, we employed the automatic plotting of $\delta$-boxplots. An example of such a plot - describing the testing scenario where the discrepancy between the reported results and our experiments was the largest, is provided in 4. Notice that, while our change in the experimental setup did contribute to slightly worse metric scores, these changes are still minimal ($\delta \in [-0.1, 0.05]$ for all observed testing scenarios). Consequentially, we conclude that the experimental results are robust for our method of risk estimation. Other diagnostic boxplots are available in the supplementary repository [Git]. The insights of Figure 4 should not be interpreted as solely positive, as we discuss in Section 5.

### 4.2.3 Alternative methods of risk estimation - Dynamical tree systems

For the Peano addition and Boolean Formulae DGPs, we attempted to employ a similar sampling restriction for training series generation, as described above. During sampling, however, we noticed that our described methodology failed to sample a sufficient amount of training examples. Our troubleshooting lead us towards the realization, that these DGPs were very prone to generating trees that could not be simplified any further, which meant that no mapping pairs $(G_0, G_1)$ could be generated from such a sample. Our diagnostic results in Figure 5 show the overwhelming majority of samples being part of this group, which casts doubt on the claims, made in 1. We evaluated the empirical probability of a unique, simplifyable tree $G_0$, being sampled from a DGP. Our results show that the Boolean addition DGP sampled such a tree with probability $\text{Pr}_{\text{Boolean}} = 0.13 \pm 0.003$, while the Peano addition DGP performed at $\text{Pr}_{\text{Peano}} = 0.26 \pm 0.002$. These results are derived over 300,000 DGP samples with uniformly distributed hyperparametric values of maximal permited operations $p \in [3, 5]$, with 3 repetitions.

In an attempt to evaluate the performance of the GEN on this family of data, we loosen our restrictions, set in Subsection 3.4. Instead, we run 5 repetitions of training, with holdout estimation (|Test set| = 100) on the time series, generated by the unique graphs $G_0$, described in the previous paragraph. In this setup, the results were not perfect, but remained in the $\pm 0.05$ standard-deviation-neighborhood of the reported results.

### 4.2.4 Performance on the social network dataset

The authors use the social network data set only to evaluate the scaling capabilities of the GEN, but do not offer any information on the model's performance on the set.[2] Since the model's scaling may be dependant on specific model parameters (specifically, the used user-defined loss function), we examine if the model is capable of training using gradient descent in this experiment. We visualize the loss curves of training the model over 1554 iterations (one pass of each available graph in Figure 6, with all hyper parameters similar to the original experiment, and using the Adam optimizer. We see that the model, implemented in the scenario, does not optimize its loss function successfully.

## 5 Discussion

Our experimental results conclusively show that most of the claims in the original work hold. It is imperative, however, to discuss the choice of DGPs on which the model was evaluated to achieve these results. Consider, for example, the Game of Life DGP, used for evaluating the precision and recall of the test set. While at first glance, a perfect result on a relatively involved system might be impressive, we must recall that node/edge edits and insertions should never appear in an edit script $\bar{\delta}$ between two Game of life graphs, as the only changes in the systems correspond to replication edits. Consequentially, the system in the scenario is only asked to output without any addition or insertion edits. Since experiment 4.2.1 showed that this output does not always appear, this casts a doubt over the model's expressive power. Another example of a somewhat poor test setup is the Edit Cycles DGP, in which the network will always test on transitions $\psi$, to which it was already introduce during training,

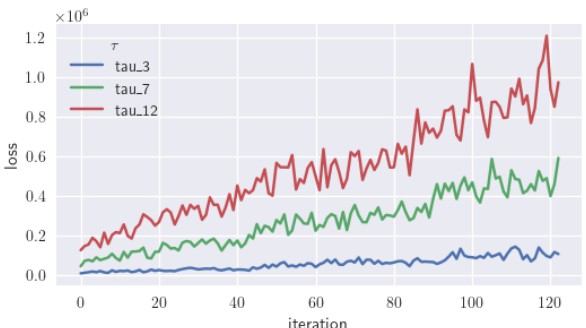

Figure 6: The loss curves for training the GEN with the authors' custom defined loss function. The differently-colored lines correspond to the values $\tau$, with respect to which the model's training set is generated.

given that the series are cyclical and Markovian. Adding to this, it is very likely that, due to the nature of the problem they describe, the mappings $\psi$, inferred from the Peano addition and Degree formulae DGPs, are often seen during training. We support this claim with our description of the sampling problems we encountered in Experiment 4.2.2.

Our experimental results on the arXiv citation network show that the network's runtimes are subquadratically dependant on the number of nodes in the given graph. This partially corroborates the authors' claims. However, we note that these results are achieved by an architecture, that is not able to optimize its loss function correctly. Given that the loss cumulative loss increases with $\tau$ (as one would expect), we hypothesize that this performance is not a result of a simple syntactical error in the author-defined loss function. While this additional insight does not disprove Claim (iii), we note that a different, better performing loss function, might.

We propose that th weakneseses we higlighted here be considered in future work, We believe that a more in-depth and practical experimental evaluation of an otherwise elegant and interpretable solution could greatly benefit the machine learning community in the years to come.

### 5.1 What was easy

All the provided code has extensive and clear documentation which made the paper's experiments easy to reproduce. The entire code base is readable, very modular, adheres to established practices on code readability, and goes hand-in-hand with the nomenclature of the paper. While the presented implementations do require intermediate familiarity with common PyTorch constructs, the authors do admirable work in explaining everything else as-they-go, almost always without using unnecessary dependencies or needlessly referencing the reader elsewhere. The authors also provide a moderate amount of clearly written unit tests for all of their models and have already pre-implemented several diagnostic

---

[2]This concern was also raised to the authors during the paper's submission and review process by *AnonReviewer4*. See the section *Weak points* in: `https://openreview.net/forum?id=dlEJsyHGeaL&noteId=Sg922s85khx`

measures, such as execution runtime logging, repetition handling and plotting of training curves, which made our work a lot easier.

## 5.2 What was difficult

Even with the extensive supplementary material, we believe that it would have been very difficult to reproduce the exact implementation of GEN and the presented DGPs by reading the paper alone, as we've discovered many important details from the supplementary documentation.

In the paper, the authors state that all experiments were run on a consumer-grade laptop. While this may be the case, running some of the provided code is prohibitively time consuming to run on such a machine. For example, we were not able to finish a single pairwise distance calculation in a day's worth of computing time (and have thus not reported on the results of that method here) on the kernel-based baseline from [PGH18]. The lack of transparency about the code base's runtimes made our work here much more difficult.

The original paper also uses a direct implementation of the [HHD$^+$19] as a baseline for experiments, relating to Claim (i). This model was not provided in the repository at the beginning of our work. The authors later provided us with the implementation, which encountered runtime errors. Even though the model now works, the trouble-shooting of this part of the code was especially time-consuming. The author's repository also lacks a hierarchical structure of related items. While the purpose of every file is clearly explained, our reproduction would have been easier with some reorganization.

## 5.3 Communication with original authors

We contacted Mr. Paaßen, along with his colleagues to inform them about our efforts to reproduce their work in mid-January. He was prompt with his responses, welcomed our work and made himself available for any questions. Upon our request, he forwarded the code with which the authors evaluated a baseline, reported in the paper, but not available in the repository. Upon our discovering of its aforementioned problems, he was prompt to offer solutions and sent us an adapted file in a couple of days. He let us know that the authors plan to update the repository with this working file shortly, which we see as an aditional benefit of our effort reproducing this article.

When asked about their method of risk estimation, the author argued that the combinatorial explosion of possible starting states makes it unlikely that GEN just memorizes the training data without generalization. For the case of the Edit Cycles dataset, where this obviously has to happen, since there is no underlying ground-truth function $\psi$, he offered the insight that generalization was not the main aim of the inclusion of this dataset. Rather, it was intended to test the expresiveness of the edits, as memoriztation alone does not suffice to solve the task of mapping $G_t \rightarrow G_{t+1}$.

Summing up, we greatly appreciate the authors' responses and their general attitude towards their work being reproduced as a part of this challenge.

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
