# OpenReview forum: "[Re] Graph Edit Networks"
_ML_Reproducibility_Challenge/2021/Fall — RC2021_

### Official Review · Reviewer_V4aJ · 2022-03-06
**Graph editing tool**

**Rating:** 6
**Confidence:** 3

**Review:**

This paper seems to follow the guidelines for reproducibility, notation, code.

The authors introduce a simple output layer (called "GEN") that can be used in graph neural networks to to obtain a means of predicting how a series of graphs will evolve over time.  The domain is graph editing.

The authors indicate GENs can be used to do several things such as a solution to the GED problem and the authors show GENs in experimental environments which are encouraging for predicting node insertions and edge operations.
The manuscript reveals that with a graph matching a pair of graphs, there is an algorithmically determinable near-optimal graph edit sequence for generating training data. Finally, the paper demonstrates the model's capabilities on a set of synthetic benchmarks.

The manuscript is well written and this paper is likely of interest to the ICLR forum broadly speaking.  This would be suitable for poster presentation.

---

### Official Review · Reviewer_f2MN · 2022-03-28
**Good work - in sync with the goal of RC**

**Rating:** 7
**Confidence:** 4

**Review:**

The authors have done a remarkable job of reproduction. The work incorporates major to-dos of RC.

---

### Meta-Review · Area_Chair_P3L4 · 2022-04-09

**Recommendation:** Accept
**Confidence:** 4

**Metareview:**

Reviewers agreed that this work should be accepted. They praised that it was a thorough reproduction, and it was well written, with good quality evaluations.

---

### Decision · Program_Chairs · 2022-04-09

**Decision:**

Accept

**Comment:**

Following the recommendation of reviewers and meta-reviewer, the paper is accepted for ML Reproducibility Challenge 2021, and will be published in the upcoming special edition of ReScience Journal.